# Novel *NDUFA13* Mutations Associated with OXPHOS Deficiency and Leigh Syndrome: A Second Family Report

**DOI:** 10.3390/genes11080855

**Published:** 2020-07-26

**Authors:** Adrián González-Quintana, Inés García-Consuegra, Amaya Belanger-Quintana, Pablo Serrano-Lorenzo, Alejandro Lucia, Alberto Blázquez, Jorge Docampo, Cristina Ugalde, María Morán, Joaquín Arenas, Miguel A. Martín

**Affiliations:** 1Laboratorio de Enfermedades Mitocondriales y Neurometabólicas, Instituto de Investigación Hospital 12 de Octubre (imas12), 28041 Madrid, Spain; agonzalez.imas12@h12o.es (A.G.-Q.); inesgcg@hotmail.com (I.G.-C.); pabloserra.lor@gmail.com (P.S.-L.); abencinar@hotmail.com (A.B.); jdocampo.imas12@h12o.es (J.D.); cugalde@h12o.es (C.U.); mmoran@h12o.es (M.M.); jarenas.imas12@h12o.es (J.A.); 2Centro de Investigación Biomédica en Red de Enfermedades Raras (CIBERER), 28029 Madrid, Spain; 3Servicio de Pediatría, Enfermedades Metabólicas Hereditarias, Hospital Universitario Ramón y Cajal, 28034 Madrid, Spain; amaya.belanger@salud.madrid.org; 4Facultad de Ciencias del Deporte, Universidad Europea, 28670 Madrid, Spain; alejandro.lucia@universidadeuropea.es

**Keywords:** mitochondrial complex I deficiency, *NDUFA13* gene, mitochondrial OXPHOS dysfunction, Leigh syndrome, OXPHOS assembly, OXPHOS diagnosis

## Abstract

Leigh syndrome (LS) usually presents as an early onset mitochondrial encephalopathy characterized by bilateral symmetric lesions in the basal ganglia and cerebral stem. More than 75 genes have been associated with this condition, including genes involved in the biogenesis of mitochondrial complex I (CI). In this study, we used a next-generation sequencing (NGS) panel to identify two novel biallelic variants in the NADH:ubiquinone oxidoreductase subunit A13 (*NDUFA13*) gene in a patient with isolated CI deficiency in skeletal muscle. Our patient, who represents the second family report with mutations in the CI NDUFA13 subunit, presented with LS lesions in brain magnetic resonance imaging, mild hypertrophic cardiomyopathy, and progressive spastic tetraparesis. This phenotype manifestation is different from that previously described in the first *NDUFA13* family, which was predominantly characterized by neurosensorial symptoms. Both in silico pathogenicity predictions and oxidative phosphorylation (OXPHOS) functional findings in patient’s skin fibroblasts (delayed cell growth, isolated CI enzyme defect, decreased basal and maximal oxygen consumption and as well as ATP production, together with markedly diminished levels of the NDUFA13 protein, CI, and respirasomes) suggest that these novel variants in the *NDUFA13* gene are the underlying cause of the CI defect, expanding the genetic heterogeneity of LS.

## 1. Introduction

Mitochondrial complex I (CI)—also known as NADH:ubiquinone reductase (H(+)-translocating, EC 7.1.1.2—is the largest of the four protein complexes that comprise the mitochondrial respiratory chain (MRC). CI includes 44 subunits [1], of which 37 are encoded by nuclear genes (*NDUFs*) and 7 are encoded by the mitochondrial DNA (mtDNA) genome (*MTNDs* genes). CI is arranged into three functional subcomplexes or ‘modules’: the catalytic N (NADH dehydrogenase), the Q (ubiquinone reduction), and the P module (proton pump) [2].

CI deficiency is one of the most frequent mitochondrial disorders in childhood [3], which is associated with mutations in several genes encoding structural CI subunits or factors involved in their assembly or stability [4]. Clinical manifestations are heterogeneous [5,6], Leigh syndrome (LS, OMIM 256000) [7] being one of the most prevalent. With onset usually before two years of age, LS is characterized at the anatomic level by bilateral symmetric lesions in the basal ganglia and cerebral stem, at the clinical level by psychomotor retardation, hypotonia, dysphonia, nystagmus, vomits, and respiratory disturbances (among other manifestations) [8,9], and at the laboratory level by elevated lactate levels in blood and cerebrospinal fluid (CSF) [10]. More than 75 genes in mtDNA or in nuclear chromosomes have been associated with LS [11]. 

The increasing use of next-generation sequencing (NGS) approaches such as targeted gene panels or whole-exome has enabled discovering new genes or mutations involved in mitochondrial oxidative phosphorylation (OXPHOS) dysfunction, including those causing LS [12,13,14,15]. In the present study, using an OXPHOS-targeted panel, we identified two biallelic novel variants in the *NDUFA13* gene in a child with features of LS and mitochondrial CI deficiency. In the patient’s fibroblasts, we demonstrated structural and functional CI defects caused by low levels of NDUFA13 protein, leading to OXPHOS dysfunction. So far, only one family (two siblings) has been reported to harbor mutations in the *NDUFA13* CI–subunit gene [16] presenting with a neurosensorial phenotype that differs from the LS shown by our patient. 

## 2. Materials and Methods 

### 2.1. Case Report

The study was approved by the Ethics Committee of the ‘Hospital Universitario 12 de Octubre’ (Madrid, Spain) and was performed in accordance with the Declaration of Helsinki for Human Research. Written consent was obtained from the parents of the patient. 

The patient was a 2-year-old boy, who was the first birth from asymptomatic non-consanguineous parents. After a normal pregnancy, in the first year of life, he showed psychomotor retardation, language skill delay, and persistent horizontal nystagmus. Brain magnetic resonance imaging (MRI) showed normal results at 6 months of age. Nystagmus improved after visual measures, but alternant strabismus was noticed. Ophthalmologic examination showed no crystalline, retinal, or optic nerve abnormalities. Physical examination revealed central hypotonia and decreased peripheral muscle tone. Lactate levels were elevated in plasma (4.0 mM, normal < 1.8) and CSF (3.0 mM, normal < 2.0). At age 2, brain MRI showed T2-weighted hyperintensity signals in basal ganglia (putamen and caudate) and cerebral pedunculus, suggesting LS. The latter MRI findings remained at age 4 and 10. At the age of 10, the patient developed mild ventricular hypertrophic cardiomyopathy (HCM). The child, who is currently 16, has acquired satisfactory cognitive skills, and presents with low weight, swallowing difficulties, spastic tetraparesis, and mild HCM. 

Muscle biopsy at age of 2 showed lipid droplets in type I fibers with no ragged-red or cytochrome c oxidase negative fibers. Assessment of muscle respiratory chain complexes and citrate synthase revealed a single CI activity (NADH–decylubiquinone–oxidoreductase) defect (Table 1). The screening of 19 frequent mtDNA mutations including two LS-associated in the *MTND5*-CI gene (e.g., m.13513G>A and m.13514A>G) failed to detect any mutation [17].

### 2.2. Molecular Genetic Studies

#### 2.2.1. NGS Studies

Deep NGS of whole mitochondrial genome was performed by Long-Range polymerase chain reaction (PCR) amplification of the entire mtDNA following protocols based in Ion-PGM sequencer (Life Technologies, Carlsbad, CA, USA). Bioinformatic analysis was carried out using a pipeline integrating variant calling format (VCF) files with Mitomap database and MitImpact predictors using in-house scripts. The mtDNA haplogroup was predicted by the Haplogrep2 platform [18].

A customized “NGS OXPHOS” panel was developed in our laboratory with the aim to identify mutations and novel OXPHOS disease-causing genes in patients with single respiratory chain complexes defects in skeletal muscle. The panel includes 133 genes (Appendix A) encoding OXPHOS structural subunits and assembly factors. The panel was designed with an Ion Ampliseq Designer v1.2 (Life Technologies), and NGS was performed with the Ion-PGM platform (Life Technologies,). Sequence alignment (ref. GRCh37/hg19) and variant detection was performed in Torrent Suite (TMAP-variant Caller plugin). The annotation and prioritization of variants was done through the integration of own scripts with Annovar [19]. Variant prioritization was based assuming an autosomal recessive inheritance following the next steps: (1) status of the variants in the ClinVar database using the InterVar tool; (2) minor allele frequency (MAF) < 0.01 in population databases such as the Genome Aggregation Database (gnomAD, https://gnomad.broadinstitute.org) and 1000 genomes project database (http://browser.1000genomes.org); (3) variant pathogenicity predictors including SIFT (http://sift.jcvi.org), PolyPhen-2 (http://genetics.bwh.harvard.edu/pph2), LRT (http://www.genetics.wustl.edu/jflab/lrt_query.html), MutationTaster (http://www.mutationtaster.org), M-CAP (http://bejerano.stanford.edu/mcap/), PROVEAN (http://provean.jcvi.org/index.php), and CADD Phred (http://cadd.gs.washington.edu), and splicing predictors such as dbNSFP (https://sites.google.com/site/jpopgen/dbNSFP), dPSI (difference in percentage spliced in) and Human Splicing Finder (http://www.umd.be/HSF3/); and (4) assessment of phylogenetic conservation using Genomic Evolutionary Rate Profiling (GERP) and the Phylogenetic Analysis with Space/Time models (PHAST) programs: phastCons and phyloP.

#### 2.2.2. Confirmation and Segregation Analysis

Sanger sequencing was performed in the proband and his parents to confirm the presence and segregation of the candidate variants in the *NDUFA13* gene. Exons 2 and 3 of the *NDUFA13* gene were amplified with specific primers by conventional PCR (Appendix A). PCR products were purified using Illustra GFX PCR DNA and a gel purification kit (GE-Healthcare, Amersham, UK) followed by Sanger sequencing in a 3130xl Genetic Analyzer (Applied Biosystems, Carlsbad, CA, USA). 

### 2.3. Cultured Skin Fibroblasts and Cell Growth Rate

Proband and control’s skin fibroblasts were cultured in Dulbecco Modified Eagle Medium (DMEM) (Lonza, Belgium) 4.5 g/L glucose supplemented with L-glutamine, 10% fetal bovine serum (Biowest, Riverside, MO, USA) and 1% penicillin/streptomycin (Gibco, ThermoFisher Scientific, Carlsbad, CA, USA). Cultures were maintained at 37 °C and 5% CO_2_ atmosphere following standard procedures. Cell growth was evaluated by seeding 25,000 control and patient’s cells per well in multiwell plates and counting at 24, 48, 72, and 96 h, respectively, as well as after 7 days. Cells were detached with trypsin and counted in the TC20 Automated Cell Counter (Bio-Rad Laboratories Inc., Hercules, CA, USA).

### 2.4. Mitochondrial Respiratory Chain Activities in Fibroblasts

Activities of the respiratory chain complexes in fibroblasts were measured spectrophotometrically using the method of Medja et al. [20] with minor modifications. Three 175 cm^2^ flasks of confluent cultured fibroblasts from the proband and control were collected by centrifugation at 800× *g* for 5 min. Pellets were resuspended in 300 to 400 µL homogenization buffer (Mannitol Buffer, pH 7.2; 225 mM Mannitol, Sigma; 75 mM sucrose, Sigma; 10 mM Tris-HCl; 0.1 mM EDTA, Sigma) and sonicated twice for 3 s in an ice bath. Fibroblast homogenates were maintained in the ice bath prior to the spectrophotometric enzyme assays.

### 2.5. Mitochondrial Respiration Assays in Fibroblasts

The oxygen consumption rate (OCR), maximal respiration rate (MRR), and ATP synthesis rate were assessed in the extracellular analyzer XFp (Seahorse Agilent Technologies, Santa Clara, CA, USA) using 20,000 cells per well from cultured skin fibroblasts derived from the proband or from a pediatric control following a protocol described elsewhere [21] with minor modifications in reagents’ final concentrations: 2.6 μM oligomycin, 1.0 μM carbonyl cyanide 4-(trifluoromethoxy) phenylhy-drazone (FCCP), and 1.0 μM rotenone/antimycin A.

### 2.6. mRNA Analysis Cultured Fibroblasts

RNA was extracted from cultured fibroblasts using TRIzol Reagent, and 2 ng RNA was retrotranscribed with SuperScript IV kit (Invitrogen, Waltham, MA, USA). The relative levels of *NDUFA13* mRNAs were determined with the SsoFast SYBR Green Supermix (Bio-Rad Laboratories Inc., Hercules, CA, USA) on a CFX96 Real-Time PCR system (Bio-Rad Laboratories Inc.). Specific primers were designed for the *NDUFA13* gene with OligoAnalyzer and PrimerQuest (Integrated DNA Technologies, Coralville, IA, USA): *NDUFA13* forward 5′-CGCCTACAAATCGAGGACTT and *NDUF13* reverse 5′-CTCCCGAAGCATCTGCAA. HPRT1 (hypoxanthine phosphoribosyltransferase 1) was used as a constitutively expressed housekeeping control gene (forward 5′-CCTGGCGTCGTGATTAGTGA and reverse 5′-CGAGCAAGACGTTCAGTCCT). mRNA quantification analyses were performed as described [22].

### 2.7. Western Blot: Assessment of NDUF13 Protein Steady-State Levels in Cultured Fibroblast

Cellular pellets were resuspended in lysis buffer with a protease inhibitor (Sigma). Protein concentration was determined by the DC method (Bio-Rad) after sonication for 15 s. Samples were incubated for 5 min in TC 4X buffer at 95 °C. Cell lysates (40 µg) were electrophoresed by SDS-PAGE in 8–16% polyacrylamide gels (Bio-Rad) and then transferred to nitrocellulose membranes (Bio-Rad) by ultra-fast 3 min transference followed by 1 h of blocking incubation (skimmed milk in 0.1% TBS-Tween). We used the following primary antibodies (Abcam, Cambridge, UK): anti-GRIM-19 (1:1000), anti-NDUFA9 (1:1000), anti-SDHA (1:10,000), anti-Core 2 (1:2000), and anti-β-Actin (1:25,000). The secondary antibody was anti-mouse IgG horseradish peroxidase (1:4000, Cell Signaling Technologies, Danvers, MA, USA). Immunoreactive bands were detected by ECL^TM^ Prime Reagent (GE-Healthcare; Amersham, UK) in an ImageQuant^TM^ LAS 4000 (GE-Healthcare, Amersham, UK) and quantified using the ImageJ software v1.8 (Wayne Rasband, NIH, Bethesda, Washington D.C., MD, USA).

### 2.8. Mitochondrial Supercomplexes Analysis and Complex I In-Gel Activity Assay

Mitochondrial-enriched fractions were obtained from cultured fibroblasts using 4 mg/mL digitonin in phosphate-buffered saline (PBS) followed by two steps of cold PBS wash and centrifugation at 13,000 rpm. To prepare mitochondrial native proteins, pellets were solubilized in 100 to 200 μL of buffer composed of 1.5 M aminocaproic acid and 50 mM Bis-Tris, at pH 7.0. After optimizing the solubilizing conditions, digitonin was used at a concentration of 4 g/g of protein. Evaluation of the steady-state levels of the MRC enzymatic complexes was carried out by blue native electrophoresis (BNE), in one (1D) or two dimensions (2D), and mitochondrial complex I in-gel activity (CI-IGA) assay, following previously described protocols [23]. Proteins were transferred to nitrocellulose membranes, and the following antibodies were used for immunodetection (Abcam): anti-NDUFA9 (1:2000), anti-SDHA (1:10,000), anti-Core 2 (1:2000), and anti-COX5A (1:1000). Secondary antibodies conjugated to horseradish peroxidase (Cell Signaling Technologies) were used, and the reactions were developed with ECL^TM^ Prime Reagent (GE-Healthcare) in an ImageQuantTM LAS 4000 (GE-Healthcare). The optical densities of the immunoreactive bands were measured by using the ImageJ software v1.8.

### 2.9. Statistical Analisis

The Student’s *t*-test (two-tailed) or Mann–Whitney *U*-test (two-tailed) was used for unpaired comparisons. Data were represented as mean ± standard deviation (SD), and GraphPad Prism 7 software was used for presentation.

## 3. Results

### 3.1. Molecular Genetics

NGS analysis of the proband’s muscle mtDNA only detected homoplasmic polymorphisms, predicting that the patient belongs to the European mitochondrial haplogroup H1.

Using a customized “OXPHOS panel”, 66 variants were called, but none was categorized in ClinVar or InterVar as pathogenic. Variant filtering by MAF <0.01 in population databases followed by the exclusion of synonymous and ‘deep intronic’ variants resulted in 12 heterozygous variants. Of these, only two were located in the same gene encoding a CI structural subunit, *NDUFA13* (NM_015965.7): (1) the missense variant c.107T>C; p.(Leu36Pro) in exon 2 of *NDUFA13,* and (2) a microdeletion c.194delT in exon 3 of *NDUFA13* that predicts a frameshift mutation leading to a premature termination codon (PTC) at the 99-residue of the protein p.(Phe65Serfs*34) (Figure 1a). Both variants were also absent in population databases such as gnomAD, 1000 Genomes project, and Spanish Variant Server (http://csvs.babelomics.org) comprising Whole Exome Sequencing (WES) and Whole Genome Sequencing (WGS) variants from 2027 individuals of Spanish ethnic background. Sanger sequencing of *NDUFA13* exons 2 and 3 confirmed the presence of both mutations in the proband´s muscle DNA (Figure 1b). p.(Leu36Pro) and p.(Phe65Serfs*34) were identified in heterozygosity in blood DNA from the father and mother, respectively, indicating that these mutations were biallelic and that the pattern of inheritance was autosomal recessive, since both parents were asymptomatic (Figure 1b). Following the American College of Medical Genetics and Genomics (ACMG) guidelines for the interpretation of sequence variants [24], both variants were classified as of uncertain significance (VUS). 

### 3.2. Biochemical and Cellular Studies in Cultured Skin Fibroblasts

To gain insight into the pathogenicity of the two novel variants in the *NDUFA13* gene that we report here, several effects on mitochondrial OXPHOS functionality were assessed in the patient’s cultured skin fibroblasts. Age-matched human cultured skin fibroblasts were used as controls [27]. 

MRC enzyme activities revealed a considerably lower level (by 65%) in CI activity relative to controls. The remainder of MRC complexes showed normal activity levels (Table 1).

The patient’s cultured fibroblasts respirometry results showed lower levels of basal OCR (by 24%) and MRR (by 47%) than the control fibroblasts, indicating decay in the electron flux through the MRC. Additionally, patient’s fibroblasts presented a lower level (by 33%) of ATP synthesis rate than control fibroblasts (Figure 2a, b). Since day 4, the cell growth rate was slower in the patient’s cells, which was reflected by a considerably lower number of viable cells at day 7 compared to the control (Figure 2c).

*NDUFA13* cDNA levels analyzed by quantitative reverse transcript-polymerase chain reaction (q-RT-PCR) were significantly decreased in the patient’s fibroblasts (approximately 45% of control), which was possibly due to the frameshift mutation (Figure 3a). Western blot showed a considerably lower NDUFA13 protein content compared to control fibroblasts. Similar results were obtained for NDUFA9, another complex I subunit, whereas the complex II and III subunits SDHA and UQCRC2, respectively, showed band intensities similar to the control (Figure 3b).

The steady-state levels of mitochondrial respiratory chain complexes and supercomplexes (SCs) or respirasomes (SC I+III_2_+IV_1–2_) were studied by blue native electrophoresis (BNE). Both CI-IGA assays and immunoblotting of one-dimension (1D) BNE membranes incubated with an anti-NDUFA9 (CI subunit) antibody revealed much lower levels (by 75%) of the CI containing SCs (i.e., SC I+III_2_ and SC I+III_2_+IV_1–2_) in the mutant fibroblasts than in the control (Figure 3b,c). Incubation with anti-UQCRC2 (CIII subunit) and anti-COX5A (CV subunit) antibodies confirmed the lower amounts of respirasomes and SC I+III_2_ in parallel with higher levels of SC III_2_+IV, dimeric CIII (CIII_2_), and free CIV in the patients’ fibroblasts (Figure 3c). The second-dimension BNE analysis (2D-BN/SDS-PAGE) again exhibited alterations in the assembly pattern of the different complexes and SCs with remarkably lower respirasome levels in the patient coupled with the accumulation of SC III_2_+IV, CIII_2_, and free CIV (Figure 3d).

## 4. Discussion

We report the case of an early-onset Leigh syndrome (<2 years) in a boy presenting with hypotonia, nystagmus, bilateral lesions in basal ganglia and lactic acidosis, and with a moderate progression of neurological symptoms. The patient, who is currently 16 years old, has developed spastic tetraparesis with swallowing difficulties, low weight, and mild HCM. Skeletal muscle biopsy showed an isolated CI activity defect and lipid droplets that prompted us to search for the genetic cause of the disorder in OXPHOS system-related genes. After discarding mutations in mtDNA, we used an NGS customized panel containing genes encoding structural subunits and assembly factors of the OXPHOS system to identify two novel heterozygous variants: a missense c.107T>C; p.(Leu36Pro) and a frameshift c.194delT; p.(Phe65Serfs*34), in the *NDUFA13* gene that encodes a structural protein of mitochondrial CI. In silico analysis of the pathogenicity for the p.(Leu36Pro) variant showed that the Leu36 residue is located within an evolutionarily highly conserved region of the NDUFA13 protein (Figure 1c). Proline is an α helix breaker and thus, the leucine to proline change could disrupt a proper α helix formation. Predictors such as SIFT, Pholyphen, Mutation Taster, CADDphred, GERP, and PhyloP indicated the damaging nature of the variant (Figure 1a). The microdeletion c.194delT predicts the generation of a PTC at residue 99, suggesting the synthesis of a truncated protein or alternatively a nonsense mRNA mediated decay event. Both variants were absent in population databases such as gnomAD and 1000 genomes. Experimental evidence for these predictions first came from the presence of a single CI activity defect (Table 1) together with decreased *NDUFA13* cDNA levels (approximately 45% of control), which may be possible due to the frameshift variant, as well as marked decreased levels of CI subunits NDUFA13 (approximately 90% of control) and NDUFA9 (approximately 85% of control) in the patient’s fibroblasts (Figure 3a). These data suggest alterations in the middle stages of CI assembly, likely affecting the junction between the CI membrane and peripheral arms [28]. 

The *NDUFA13* gene, also known as GRIM-19, was initially reported to be involved in apoptosis induction in the nucleus by the interferon β/retinoic acid pathway [29]. More recently, it was identified as a constituent of bovine mitochondrial CI [30]. Several studies have demonstrated that somatic *NDUFA13* mutations favor tumorigenesis in different cancers, with this gene thus acting as tumor-suppressor gene [31]. The NDUFA13 subunit is located in the P module of CI, which is an essential site for the stabilization of N and Q modules and thus for the complete assembly of CI. In human-cultured cells, CI seems to be found exclusively as a component of the respiratory supercomplexes (i.e., the respirasome and SC I+III_2_) and requires CIII for its full assembly [32,33]. NDUFA13 dysfunction leads to a defect in the interaction and stability of other structural subunits belonging to the different modules of the CI holoenzyme and, as a consequence, to an altered formation of SCs [34]. Consistently, the patient’s mutant fibroblasts showed a severe complex I deficiency (by 35% compared to control) confirmed by a marked decline in the activity of the respirasome in CI-IGA assay. The reported CI deficiency could be the result of an unstable NDUFA13 aberrant protein, leading to lower respirasome and SC I+III_2_ levels. Thus, CI deficiency resulting from the disruption of complex I assembly/stabilization in the SCs could explain the delayed growth rate found in patient’s fibroblasts, as well as the significantly lower levels of basal OCR, MRR, and ATP production rate. 

Complex I deficiencies cause a wide spectrum of clinical phenotypes, presenting predominantly as LS, encephalopathy, neonatal HCM, leukodystrophy, and fatal infantile lactic acidosis [4]. Moreover, they are genetically heterogeneous, since they have been associated with mutations over 20 genes encoding assembly or structural subunits of nuclear origin [35] and in the seven CI subunits mtDNA-encoded genes [36], making diagnosis challenging. In our case, by the use of a customized NGS panel, we eventually achieved the genetic diagnosis several years after the symptom onset [15], supporting the relevance of NGS approaches not only to reach accurate molecular diagnosis in rare genetic disorders, including OXPHOS diseases [37], but also to detect novel mutations even in an highly heterogeneous and exhaustively studied condition such as LS (i.e., with more than 75 different genes potentially implicated [11]).

To our best knowledge, only one family has been previously reported to carry a mutation in the *NDUFA13* gene associated with CI inherited defects [16]. Two affected sisters harbored a homozygous missense mutation p.(Arg57His). The eldest sister was a 12-year-old when diagnosed and showed severe hypotonia, dyskinesia, lactic acidosis, and sensorineural defects (hearing loss, optic neuropathy, and retinal dysfunction). By contrast, our patient showed an early-onset LS (hypotonia along with bilateral basal ganglia lesions and lactic acidosis) with a moderate progression of neurological symptoms, and he is alive at age 16. He has also mild HCM and developed spastic tetraparesis but to date, no sensorineural signs have been documented. Establishing the reason our patient presents a different phenotype of that of the first family is not straightforward, as LS is a mitochondrial disorder with phenotype and genotype heterogeneity. The differences in clinical expression with respect to the first reported cases could be explained at least in part by the distinct findings in the OXPHOS function. Although some results from OXPHOS functionality experiments were similar in both families, we particularly found very reduced levels of the NDUFA13 protein in the patient’s fibroblasts (90% of control), while the first family had a less severe decrease (mean reduction in both sisters was 70% of control). Alternatively, factors other than OXPHOS dysfunction may influence the differential phenotype expression [38]. Our results from SC assembly studies in cultured fibroblasts are similar to those found in the first family reported (i.e., loss of CI stability and diminished formation of the respirasome [I+III_2_+IV_1–2_]), which supports that the NDUFA13 protein might play a critical role in the assembly or stability of the respirasomes, as previously shown in *NDUFA13* knockout HEK293T cells [39]. Furthermore, we have shown in our patient that the novel biallelic mutations lead to decreased mitochondrial respiration and ATP production, which could explain the decreased cell growth rate.

CI deficiencies caused by mutations in the structural assembly of CI nuclear genes represent a heterogeneous group of disorders that are frequently associated with premature mortality [36]. Our patient and the previously reported sisters [16] had early-onset symptoms and a non-rapid clinical evolution—which is consistent with the fact that the patients are alive in the second decade of life, suggesting that *NDUFA13* mutations could have less severe functional consequences than other mutations associated with CI deficiencies or with LS [8]. On the other hand, since only three patients (two unrelated families) have been documented so far and severe OXPHOS dysfunction was observed in the patient’s cells studied here, it is highly likely that other factors influence the clinical progression of the disease. Regarding the relatively benign clinical outcome of *NDUFA13* mutations, our results suggest that alternative OXPHOS-unrelated functions of this gene could be important in defining the characteristics of the clinical phenotype.

To sum up, we report on a patient from the second family with novel pathogenic variants in the NDUFA13 subunit of mitochondrial CI accounting for LS, which is a clinical phenotype that is different from the sensorineural symptoms described in the first family. Our results suggest that NGS is a valuable tool to reach genetic diagnosis even years after the initial symptoms of a typical OXPHOS disorder, such as LS and confirm—along with the findings in the previous family reported [16]—that biallelic mutations in the pleiotropic *NDUFA13* gene might be lead to less severe consequences than other lethal mutations in CI nuclear genes. Likewise, our work highlights the need to carry out functional studies in patients’ tissues and cells when VUS are prioritized during genetic analysis and they are consistent with clinical, biochemical, and inheritance features, aiming to change the variant ACMG classification to “pathogenic” or “likely pathogenic”.

## Figures and Tables

**Figure 1 genes-11-00855-f001:**
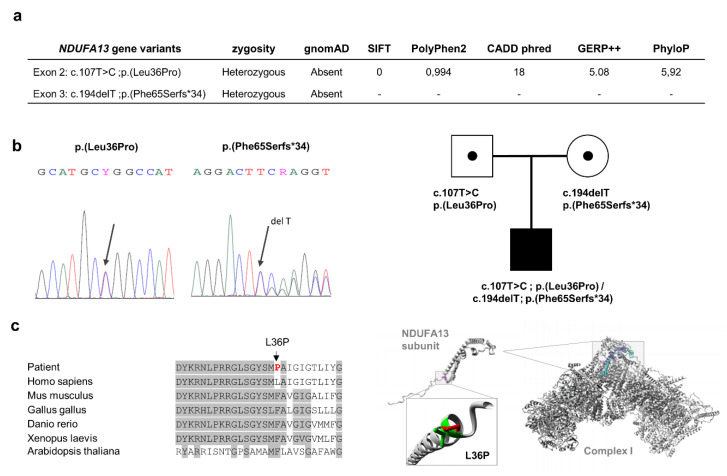
*Variant analysis of the NDUFA13 gene*. Panel (**a**) indicating the two prioritized variants in the *NDUFA13* gene and some predictor scores, after the “OXPHOS panel” analysis. Panel (**b**) showing Sanger sequencing and family segregation of both variants. Panel (**c**) left side, denoting the evolutionary conservation region (gray color) where amino acid residue L36 is located in the *NDUFA13* gene; red color, mutant proline residue; right side, structural analysis of p.(L36P) mutation using the HOPE program [25], representing the overlay of the wild-type and mutant (green and red, respectively) residues of the human NDUFA13 protein; the mutant residue (P) is smaller than the wild-type residue (L), while a destabilizing effect of proline was described in the α helix structures [26], which might disrupt protein–protein interactions with other structural subunits to either assemble or stabilize complex I. OXPHOS: oxidative phosphorylation.

**Figure 2 genes-11-00855-f002:**
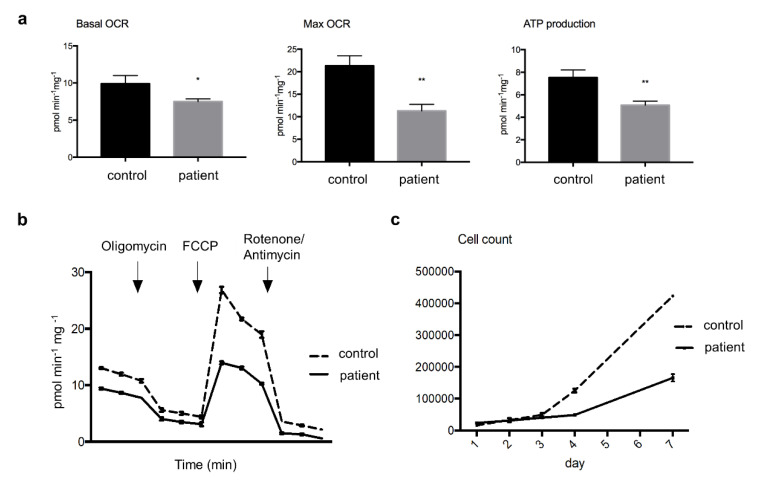
*Oxygen consumption rates (OCR) and cell growth in cultured skin fibroblast*. Panel (**a**) showing significantly lower levels of basal and maximal OCR (Mann–Whitney *U*-test, * *p* < 0.05, and ** *p* < 0.01, respectively), as well as of ATP production rate (** *p* < 0.01) in the patient’s compared to control cells; Panel (**b**) a representative three-replicate analysis of OCR in an XF extracellular analyzer; Panel (**c**) displaying slower cell growth rate in patient’s cultured cells compared to control assayed for one week. Abbreviation: FCCP, carbonyl cyanide-**4**-(trifluoromethoxy) phenylhydrazone.

**Figure 3 genes-11-00855-f003:**
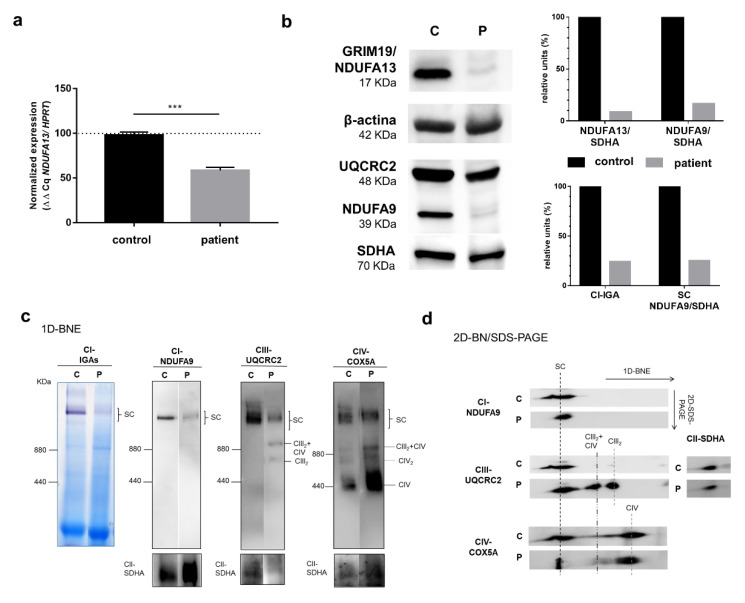
Analyses of NDUFA13 mRNA expression, NDUFA13 protein levels, and structural state of OXPHOS supercomplexes (SC) in cultured skin fibroblasts. Panel (**a**) NDUFA13 mRNA expression levels by quantitative q-RT-PCR on mRNA isolated from cultured skin fibroblasts of a control and the patient. Results were normalized to hypoxanthine phosphoribosyltransferase 1 (HPRT) mRNA levels and expressed as percentage of control; n = 3, error = SD. p-values were calculated by unpaired t-test. *** p < 0.001. Panel (**b**) Western blot showing lower levels of complex I subunits NDUFA13 and NDUFA9, in the patient’s cells compared to the control; upper right diagram, Western blot densitometric values for NDUFA13 and NDUFA9 normalized by SDHA (CII); lower right diagram, densitometric values of CI-IGA and 1D-BNE (CI-NDUFA9) normalized by SDHA (CII). Panel (**c**) Complex I in-gel activity (CI-IGA) assay and one-dimensional blue native electrophoresis (1D-BNE) analysis of digitonin-solubilized mitochondria from control and proband’s fibroblasts showing OXPHOS complexes I to IV (CI–CIV). Complex II (anti-SDHA) was used as a loading control. Ferritin was used as molecular mass standard (on the left of gels): monomer (440 kDa) and dimer (880 kDa); Panel (**d**) 2D-BN/SDS-PAGE analysis of the steady-state levels of respirasome and SC I+III_2_ (SC). CIII_2_+CIV, supercomplex containing CIII_2_ and CIV. CIII_2_, dimeric CIII. CIV_2_, CIV dimer. C: control, P: patient.

**Table 1 genes-11-00855-t001:** Mitochondrial respiratory chain enzyme activities in skeletal muscle and cultured skin fibroblasts.

	Skeletal Muscle		Fibroblasts	
	Patient (Controls) ^1^	% ^2^	Patient (Controls) ^3^	% ^4^
CI (NADH–DB oxidoreductase)	8.1 (11.3–24.7)	72%	7.1 (20.4 ± 2.3)	35%
CII (Succinate dehydrogenase)	4.7 (5.8–19.9)	81%	58.5 (48.4 ± 7.16)	120%
CIII (DBH_2_–Cytochrome c oxidoreductase)	52.3 (31–127)	169%	162 (160 ± 22)	101%
CIV (Cytochrome c oxidase)	21.1 (20–79.2)	106%	66.2 (55.4 ± 14.3)	119%
CS (Citrate synthase) ^5^	212 (102–257)	-	47.1 (77.0 ± 27.0)	-

(1, 3) nmol min^−1^ mg^−1^ protein enzyme activity expressed with respect to citrate synthase (CS) activity. (1) Control range, 2.5th–97.5th percentile (n = 95). (2) % of complex activity relative to the 2.5th percentile of controls. (3) Control mean ± SD (n = 3 cultured skin fibroblasts assayed in parallel to patient´s fibroblasts). (4) % of complex activity relative to the mean control values. (5) CS activity expressed as nmol min^−1^ mg^−1^ protein.

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
