# Peer review of "Novel NDUFA13 Mutations Associated with OXPHOS Deficiency and Leigh Syndrome: A Second Family Report"

_genes, 2020, doi:10.3390/genes11080855_

Round 1

Reviewer 1 Report

The authors describe novel potentially disease-causing mutations in the NDUFA13 gene encoding a complex I (CI) subunit. As a whole, the report is not very convincing. CI comprises 44 subunits and its assembly is controlled by more than a dozen known assembly factors. Moreover, some CI subunits are degraded when CI assembly is compromised, and this is the case for NDUFA13 which is often decreased when CI assembly is abolished. Therefore, a decrease of NDUFA13 protein on an immunoblot is hardly a strong evidence that this is the disease-causing gene. Sure, there is one frameshift allele, but the other allele carries a missense mutation of a poorly conserved amino acid and may encode a functional protein. The article would have been much more convincing if the authors have attempted to complement the defects observed in patient fibroblasts by expressing wild type NDUFA13, which is the standard (and scientifically sound) approach for reports of novel mutations presenting with new symptoms. To clarify, I am not questioning if the patient has a CI-induced LS, but I am not convinced that it is caused by the mutations in NDUFA13.

Two minor comments:

  1. Regarding the segregation shown in figure 1B: -/- and +/- are formally used to indicate homozygous loss of function allele for a specific gene or heterozygosity with a wild type allele rather than presence or absence of a specific variant. These should be removed. There are plenty of examples in the literature how to best present this information.
  2. To my knowledge, mitochondrial dysfunction rarely, if at all, results in growth defects of fibroblats in glucose-rich media. Is this something specific to fibroblasts with CI defects or could it indicate a problem with the fibroblast line itself? A complementation assay should show if the growth defect is indeed caused by NDUFA13 and CI deficiency.

Reviewer 2 Report

Using NGS the authors identified two heterozygous variants in NDUFA13 in trans.  Since both variants were novel, the authors applied standard biochemical approaches to prove the pathogenecity of the variants.  The study is solid.  All methods are well established standard approaches.  This is the second case of NDUFA13.  This patient's clinical expression differed from t hat of the first reported case.

The authors should discuss and speculate the mechanisms that lead to the differential phenotype expression. 

Round 2

Reviewer 1 Report

Despite the presence of some improvements in the manuscript, the main question “Is NDUFA13 the disease-causing gene in the patient?” remains unanswered. Instead of providing evidence by carrying out the only experiment that would demonstrate the role of NDUFA13 in the observed molecular phenotypes, the authors list 8 arguments to support their believe in the pathogenicity of the mutations. Some arguments are either computational/database analyses and/or rely on the preexisting assumption that NDUFA13 is the disease-causing gene; the argument by the authors that they have excluded all 13 assembly factors can be countered by noting that there are more than 13 CI assembly factors. Finally, the arguments that the patient and the fibroblasts have isolated CI defect do not in any way point specifically to NDUFA13.